# Neuronal Plasticity and Age-Related Functional Decline in the Motor Cortex

**DOI:** 10.3390/cells12172142

**Published:** 2023-08-25

**Authors:** Ritsuko Inoue, Hiroshi Nishimune

**Affiliations:** 1Laboratory of Neurobiology of Aging, Tokyo Metropolitan Institute for Geriatrics and Gerontology, 35-2 Sakaecho, Itabashi-ku, Tokyo 173-0015, Japan; inoritu@tmig.or.jp; 2Department of Applied Biological Science, Tokyo University of Agriculture and Technology, 3-8-1 Harumicho, Fuchu-shi, Tokyo 183-8538, Japan

**Keywords:** aging, CoQ_10_, long-term potentiation, LTP, mitochondria, motor cortex, ubiquinone

## Abstract

Physiological aging causes a decline of motor function due to impairment of motor cortex function, losses of motor neurons and neuromuscular junctions, sarcopenia, and frailty. There is increasing evidence suggesting that the changes in motor function start earlier in the middle-aged stage. The mechanism underlining the middle-aged decline in motor function seems to relate to the central nervous system rather than the peripheral neuromuscular system. The motor cortex is one of the responsible central nervous systems for coordinating and learning motor functions. The neuronal circuits in the motor cortex show plasticity in response to motor learning, including LTP. This motor cortex plasticity seems important for the intervention method mechanisms that revert the age-related decline of motor function. This review will focus on recent findings on the role of plasticity in the motor cortex for motor function and age-related changes. The review will also introduce our recent identification of an age-related decline of neuronal activity in the primary motor cortex of middle-aged *mice* using electrophysiological recordings of brain slices.

## 1. Introduction

Voluntary activation of skeletal muscle tends to weaken with age, particularly in the elderly with sarcopenia or frailty [1,2]. Physiological aging causes a decline of motor function due to losses of muscle mass and strength, denervation of neuromuscular junctions (NMJs), a loss of motor neurons in the spinal cord, and impairment of motor cortex function. Losses of muscle mass and strength (sarcopenia) are more pronounced in the elderly [3,4,5,6,7]. Meanwhile, a decline in motor function has been reported in studies of middle-aged *humans* and rodents. Middle-aged healthy subjects (between the late 40s and late 50s) changed their kinetic characteristics of gait during walking and running [8]. The age-related decline of motor function has been reported in middle-aged rodents (between approximately 13 months and 19 months old) using behavioral tests that assess balance and motor coordination or gait [9,10,11]. These data suggest that a decline in motor function in rodents starts much earlier than a decline in survival rates, which is typically around 24 months old. In addition, the mechanism of motor function decline during the middle-aged stage seems different from that during the advanced-aged stage. This difference is suggested because NMJ denervation, motor neuron loss, and muscle atrophy are detected in the advanced-age stage, but NMJ denervation is not detected significantly in middle-aged *mice* at or earlier than 18 months old [12,13]. The preservation of NMJs suggests that the spinal motor neurons are maintained in middle-aged *mice*. Furthermore, a decline in muscle contractility is less severe in *mice* under 20 months old [14].

In the motor cortex, physiological aging also causes cortical atrophy, alteration of excitability, and decreased neurotransmitter levels (Figure 1) [15,16]. Notably, age-related alteration of excitability in the motor cortex has been linked to motor function decline in *humans* and animals. In the *human* motor cortex, physiological aging alters the balance of excitatory and inhibitory circuits [17,18]. Middle-aged and elderly individuals (between the late 50s and early 70s) have exhibited more intracortical inhibition and less intracortical facilitation in the motor cortex than young adults. The hypoexcitability in the motor cortex correlates with behavioral impairments in chronic obstructive pulmonary disease (COPD) and amyotrophic lateral sclerosis (ALS) patients [19,20]. Therefore, alteration of motor cortex function may be part of the underlying mechanism for the age-related decline of motor function, especially in the middle-aged stage.

Physiological aging alters synaptic plasticity. In *rat* hippocampal slices, NMDA receptor-dependent long-term potentiation (LTP) decreases with age [21,22]. Meanwhile, neuronal plasticity enhancers have been shown to accelerate the rehabilitation-like effect that improves motor function [23,24]. Rehabilitation training restores motor function after stroke and nervous system damage. Therefore, neuronal plasticity seems to play a role in the maintenance and/or improvement of motor function. However, there has been less focus on the role of synaptic plasticity in the study of age-related decline of motor cortex function. This review highlights recent articles reporting age-related alterations of neuronal activity and synaptic plasticity efficiency in the motor cortex, which may cause a decline in motor function. We will discuss potential intervention methods by supplementing brain substances that decrease with age. These endogenous substances can directly or indirectly enhance LTP and may ameliorate the age-related alterations of motor cortex function.

## 2. Motor Cortex

### 2.1. Pathway

The motor cortex is the center of cortical control of voluntary movements [25,26]. The functional involvement of motor cortex networks in motor control has long been studied mainly by microstimulation of the rodent motor cortex [27,28]. Recently, high-precision stimulation techniques using scanning laser light stimulation and optogenetics have revealed intralaminar excitatory connections in the motor cortex [29,30,31]. These cell-selective stimulation experiments demonstrate that the excitatory connection from layers II/III to V is the major intralaminar connection in the motor cortex [29,32,33,34]. Layer V of the motor cortex contains pyramidal neurons that are projection neurons [35]. Corticospinal and cortico-brainstem projection neurons are located in the deeper layer, layer Vb. Approximately 60% of total corticospinal projections originate from the primary motor cortex (M1) in *humans* [35,36].

### 2.2. Plasticity in the Motor Cortex

#### 2.2.1. Synaptic Plasticity and Motor Learning

The motor cortex plays an essential role in motor learning [37]. Neuronal plasticity in the motor cortex during and after motor learning has been studied in *humans* and animals (Table 1) [38,39,40,41,42,43]. A rotor rod task induces glutamatergic or GABAergic synaptic plasticity in layer V pyramidal neurons of the *rat* primary motor cortex [41]. Motor learning in *mice* or stimulation of brain slices using electrodes can induce LTP in the motor cortex [39,40,44,45,46,47,48]. Changes in motor cortex excitability have a functional relationship with LTP. After LTP expression, field excitatory postsynaptic potential (fEPSP) shows enhanced amplitudes in the motor cortex [42,49,50]. A pellet-reaching task with one forelimb of *rats* increases the fEPSP amplitude of layer II/III horizontal connections in the brain slices of the trained side of the primary motor cortex [6,44]. When an LTP induction method has been applied to these slices, the amount of LTP induced in the trained contralateral side of the primary motor cortex has been less than that in the untrained ipsilateral side of the primary motor cortex in the same slice. This difference has been assumed that the trained side of the primary motor cortex had fewer LTP induction trials to reach LTP saturation, at which point LTP was no longer induced, than the untrained side of the primary motor cortex. Even under LTP saturation, LTP could be induced again if long-term depression (LTD) was induced by electrical stimulation in the motor cortex of the *rat* slice [42]. Similarly, the temporal occlusion of LTP in the *human* motor cortex is necessary to retain motor learning according to transcranial magnetic stimulation experiments [38,51,52,53]. Consequently, these studies in animals and *humans* have revealed that the mechanism of motor learning involves induction and occlusion of LTP.

Furthermore, the amount of LTP in the motor cortex is related to the success rate of motor learning. Several brain substances have been reported to affect the amount of LTP in the motor cortex. Impairment of dopaminergic signaling reduces the amount of LTP in layer II/III in the *rat* primary motor cortex, resulting in lower skill acquisition rates [55,59,63]. Blockade of Nogo-A, an inhibitor of axonal outgrowth and regeneration in the brain, enhances the amount of LTP in layer II/III in the *rat* primary motor cortex and improves motor learning [58]. These recent studies have shown that modulating neuronal plasticity in the motor cortex may alter motor learning performance.

#### 2.2.2. Cortical Plasticity Induced by Non-Invasive Stimulation

Cortical plasticity in the motor cortex has been suggested in studies of motor function in healthy subjects [64,65], motor deficits in Parkinson’s disease [59,66,67], and rehabilitation after brain damage [24,68]. These studies indicate that the induction of cortical plasticity in the motor cortex could be beneficial for motor function. Cortical plasticity can be induced by non-invasive brain stimulation techniques like transcranial magnetic stimulation and transcranial direct current stimulation (tDCS) [69]. These non-invasive brain stimulation through the scalp to the motor cortex induces motor evoked potentials (MEPs), which are recorded from the muscles. Generally, *human* cortical plasticity in the motor cortex has been monitored by changes in the amplitude of peripherally recorded MEPs [64] because it is currently difficult to directly measure long-term changes in excitability in the *human* motor cortex. Importantly, tDCS has been suggested to directly induce synaptic plasticity based on studies using cortical slices of rodents [49,70,71]. When tDCS electrodes were used to stimulate brain slices, the stimulation-induced NMDA receptor-dependent synaptic potentiation in layer II/III of the *mouse* primary motor cortex in brain slices [50]. The brain slices of tDCS-treated *mice* showed LTP enhancement at layer II/III horizontal connections of the primary motor cortex [49].

#### 2.2.3. Structural Plasticity and Motor Learning

In addition, neurons in the motor cortex show structural plasticity concurrently to the LTP induced by motor learning or experimental induction methods. In the motor cortex of *rats*, spine dynamics differ in a laminar structure and cell-dependent manner [54,61]. Interestingly, motor learning causes the stabilization of new dendritic spines [56,57,72]. Furthermore, new dendritic branches and increased dendritic length have been observed among layer V pyramidal neurons of *rats* that learned a pellet-reaching task. After completing these motor tasks, the degree of these dendritic changes peaks in one month in the motor cortex, after which these newly formed dendrites are pruned [60].

## 3. Aging in the Motor Cortex

### 3.1. Structural and Functional Alteration in the Motor Cortex

Physiological aging induces changes in the *human* motor cortex, including cortical atrophy, impaired excitability, and decreased neurotransmitter levels [15,16,73,74]. Generally, cortical atrophy in physiological aging is considered to be due to structural changes in individual neurons, such as shrinkage in the soma size, a reduction in the dendrite arborization complexity and length, and a loss or regression of dendritic spines but not a decrease in cell number [75]. These age-related structural changes occur in parallel with alterations in the electrophysiological properties of neurons. In the aged *primate* prefrontal cortex, single action potential amplitude and fall time significantly decrease in layer II/III and layer V pyramidal neurons [76,77,78]. Furthermore, action potential firing rates in layer II/III increase with age and are related to cognitive function. Subjects with intermediate firing rates demonstrate higher performance in delayed non-match to sample (DNMS) basic learning, DNMS performance at 2 min delays, and the delayed recognition span task compared to subjects with low or very high firing rates [76,79]. Age-related alterations in the electrophysiological properties impair the excitability of neural networks at the cellular level in the cortex [80].

In the motor cortex, age-related structural changes in dendritic spines occur prior to neuronal cell death. Repeated in vivo imaging of the same dendrites has revealed that dendritic spine density increases and long-term spine survival decreases in layer V pyramidal neurons of aged *mice* (20 to 22 months old) compared to young *mice* (3 to 4 months old) [81].

According to young-adult rodent studies, the excitability of layer V in the motor cortex plays an important role in motor function. Layer V pyramidal neurons in the motor cortex control the rhythm of whisker movements [82]. Interestingly, enhancing the activity of layer V neurons in the motor cortex improves bradykinesia and hypokinesia in Parkinson’s disease model *mice* [31]. *Human* studies have demonstrated the age-related decline in motor cortex excitability by measuring peripheral neural activity that indirectly represents motor cortex activity [73,83]. However, age-related changes in motor cortex activity have not been recorded directly in *humans* or animals. Importantly, we have discovered an age-related decline in field excitatory postsynaptic potential (fEPSP) amplitude in the pathway from layers II/III to V of the primary motor cortex of middle-aged *mice* (15 to 18 months old) for the first time to the best of our knowledge [50]. These middle-aged *mice* performed slower on the pole test than younger *mice* (6 months old) [10,50]. Therefore, layer V neuronal activity may be a valuable target for examining the relationship between motor cortex excitability and age-related motor function decline.

### 3.2. Aging and Cortical Plasticity in the Motor Cortex

Age-related alteration of plasticity in the motor cortex is related to age-related reduction in motor function. As mentioned above, middle-aged *mice* show behavior impairment in the pole test compared to younger *mice* [10,50]. These *mice* show age-related alteration of neuronal plasticity in the pathway from layers II/III to V of the primary motor cortex, which will be described in detail in section four [50]. Aged *mice* (18 to 20 months old) could learn the pasta matrix reaching tasks as the young *mice* (3 to 5 months old). However, the brain mapping study using intracortical microstimulation of layer V in the motor cortex has revealed age-related alteration in the motor map plasticity after the reaching task [84]. In *humans*, magnetoencephalography and functional magnetic resonance imaging (fMRI) studies have shown age-related alteration of plasticity in the motor cortex by measuring movement-related beta desynchronization during motor execution in the elderly [85]. These studies about motor learning and motor function have shown that cortical plasticity in the motor cortex differs between young and older animals.

### 3.3. Aging and Mitochondria in the Motor Cortex

It is essential to maintain adequate mitochondrial activity for producing ATP to meet the high energy requirements of neurons. ATP production correlates highly with the complex I activity in mitochondrial oxidative phosphorylation. The non-synaptic and synaptic mitochondria in the brain have different ATP production efficiency. In non-synaptic mitochondria of the brain, ATP production is not affected until the complex I activity is reduced by 72% [86]. However, ATP production is significantly reduced when the complex I activity of synaptic mitochondria of the brain is reduced by 25% [87]. These findings suggest that ATP production at cortical synapses seems sensitive to modest dysfunction of mitochondria. Interestingly, mitochondrial activity is significantly impaired in synaptic mitochondria but not in non-synaptic mitochondria during physiological aging in middle-aged rodents (14 or 17 months old) [88,89].

Mitochondria affect synaptic activities, including age-related synaptic plasticity [90,91]. In the *mouse* brain, the complex I activity in mitochondrial oxidative phosphorylation decreases with age [10,92]. We have discovered age-related alteration of synaptic plasticity efficiency in the primary motor cortex in middle-aged *mice*. LTP in the primary motor cortex in middle-aged *mice* has been enhanced by high-frequency stimulation combined with the administration of the mitochondrial coenzyme, coenzyme Q_10_ (CoQ_10_), which declines with age [50]. However, the same stimulation with CoQ_10_ administration does not affect LTP in the primary motor cortex of young-adult *mice* [50]. These studies suggest that mitochondrial dysfunction results from brain aging [93] and is part of the mechanisms that alter neuronal plasticity in the motor cortex.

## 4. Interventions for Age-Related Declines of Motor Function

Age-related declines in motor function may be caused partly by mitochondrial dysfunction. Physiological aging is known to cause mitochondrial dysfunction in skeletal muscles and NMJs of *humans* and animals. The mitochondria of *human* skeletal muscle are altered by physiological aging resulting in decreased enzyme activity, maximal respiration capacity, and total protein amount [94,95]. The NMJs of aged *rats* have megamitochondria, which is thought to be generated by age-related mitochondrial fusion [96,97]. A relationship between the mitochondria function and the motor function has been shown in exercise intervention studies, which activates mitochondrial biogenesis and improves muscle performance in *humans* and rodents [98].

Similarly, studies have reported correlations between age-related declines in brain mitochondrial function and motor function. The basal ganglia of aged *monkeys* have significantly reduced ATP production, pyruvate dehydrogenase activity, and calcium buffering capacity compared to younger animals. These reductions correlate with age-related decline in locomotor activity and movement speed [99]. Takahashi et al. have shown reductions in brain mitochondrial respiratory capacity, coenzyme Q (CoQ) content, and motor function in middle-aged *mice* compared to young *mice* [10]. CoQ is an essential mitochondrial coenzyme for ATP production [100,101,102]. Electrons are transported from complexes I and II to complex III by coenzyme Q_10_ in *humans* and coenzyme Q_9_ and Q_10_ in *mice* for ATP production [103,104,105]. However, CoQ levels decline with aging in the brain, blood, and other organs. The symptom onset of aging, CoQ_10_ deficiency, multiple-system atrophy, and Parkinson’s disease have been alleviated and/or delayed by CoQ_10_ supplementation in *humans* [106,107,108].

CoQ is a fat-soluble substance whose bioavailability varies between formulations [109,110,111]. Optimal concentrations and effective administration periods differ depending on the CoQ_10_ formulation supplemented. In a Parkinson’s disease model *mouse*, an intervention utilizing CoQ_10_ nano micelles with enhanced brain penetration capability has shown neuroprotection and improved motor function [112]. Another water-soluble nanoformula-type CoQ_10_ supplementation by drinking water has improved complex I activity in brain mitochondria, CoQ contents in the brain, and motor function in middle-aged *mice* [10,92]. We have discovered that CoQ_10_ supplementation by drinking water also improves the age-related decline of fEPSP amplitude in the pathway from layers II/III to V of the primary motor cortex of middle-aged *mice*. In addition, CoQ_10_ administration combined with high-frequency stimulation induces age-dependent LTP and enhances the basal fEPSP amplitude level in brain slices of the primary motor cortex. This LTP induction is age, CoQ_10_, and NMDA receptor-dependent [50]. Layer V of the motor cortex was thought to be a region unlikely to induce LTP compared to layer II/III of the motor cortex based on brain slice experiments [54]. However, layer V neurons in the primary motor cortex show dynamic alteration of synaptic plasticity after motor learning [41]. Stimulation of superficial layers of the motor cortex using a concentric bipolar electrode induces LTP in layer V pyramidal neurons in the M1 forelimb area [59]. Our data suggest that LTP could be induced in layer V neurons of the primary motor cortex in slice preparation under certain conditions. The CoQ_10_-dependent enhancement of NMDA receptor components in our study may be similar to the ability of growth hormone to reverse the age-related decrease in NMDA receptor function in basal excitatory transmission. This is because chronic growth hormone treatment restores NMDA receptor-dependent basal neuronal activity in aged *rat* hippocampus [113]. The LTP or related neuronal plasticity mechanism in vivo during CoQ_10_ supplementation may have boosted neuronal activity in the primary motor cortex and improved the motor function of middle-aged *mice* [50]. These findings indicate that restoration of mitochondrial function enhances neuronal activity and improves motor function, at least in middle-aged *mice*.

In addition, studies have reported correlations between CoQ_10_ and neurotrophin, brain-derived neurotrophic factor (BDNF). *Rats* exposed to chronic unpredictable mild stress or toxic propionic acid show a decline in BDNF protein levels. CoQ_10_ treatment in these *rats* results in a modest increase in BDNF protein [114,115]. BDNF is associated with synaptic plasticity in the hippocampus [116]. In the motor cortex, layer II/III and V neurons express BDNF protein. Running wheel exercise increases the number of BDNF-expressing pyramidal neurons in layer II/III and the expression level of BDNF protein in the *mice* motor cortex. However, reduced BDNF protein levels in conditional BDNF knockout *mice* impair motor learning [117], suggesting that BDNF protein is required in the motor cortex during motor learning. Fritsch et al. have suggested that direct current stimulation combined with repetitive low-frequency stimulation increases activity-dependent BDNF secretion and improves motor learning by enhancing NMDA receptor-dependent LTP at layer II/III in the primary motor cortex of *mice* [70]. These results are consistent with the study demonstrating that BDNF secretion modifies neuronal plasticity in an activity-dependent manner in the hippocampus [118]. Based on these results, intervention methods to improve motor function and learning via the effects of BDNF have been investigated in combination with stimuli such as exercise and tDCS [70,119,120]. The correlation between the amount of activity-dependent BDNF secretion and motor learning has been examined using *BDNF* valine-methionine substitution (Val66Met) polymorphism. The Val66Met polymorphism is a single nucleotide polymorphism in the *BDNF* gene related to episodic memory in *humans* and the activity-dependent secretion of BDNF [121,122]. The Val66Met substitution decreases activity-dependent secretion of BDNF in cultured *rat* hippocampal neurons [121]. Healthy subjects (mean age 22.7 ± 1.4 years) with the Met allele show reduced motor evoked potentials (MEPs) and brain motor map area changes with training using right index fingers compared to subjects without the polymorphism [123]. In both *humans* and *mice*, direct current stimulation enhances motor skill acquisition rates in Val/Val subjects without the polymorphism but not in Met allele carriers [70].

The *BDNF* mRNA expression level declines with age in the *human* prefrontal cortex [124] and the *monkey* motor cortex [125]. BDNF protein levels have affected motor function in the *mice* lacking one *BDNF* allele. These *BDNF* heterozygous *mice* at the middle-aged (11 to 13 months old) and the aged (19 to 21 months old) stages walk slower on a horizontal beam than age-matched wild-type *mice*. BDNF protein levels in the striatum of aged *BDNF* heterozygous *mice* are 15% lower compared to aged wild-type *mice* [126]. The relationship between age-related decline in BDNF levels and motor function in the elderly is not yet well understood. The serum BDNF level correlates positively with the eigenvector centrality obtained from resting-state fMRI data in the premotor and motor cortex of the elderly [127]. The eigenvector centrality analyzes connectivity patterns of the *human* brain in fMRI data [128]. Although the Val66Met polymorphism affects motor performance in young people, this effect is not well understood in the elderly. Healthy elderly subjects (mean age 73.2 ± 1.8 years) show slower reaction time speed, a larger baseline of brain motor map area, and smaller MEP amplitude compared to young subjects (mean age 24.3 ± 1.1 years) [129]. Among these healthy elderly subjects, the polymorphism difference between *BDNF* Val/Val and Val66Met has not affected reaction time speed, the motor tasks using hands or fingers, driving test of cognitive/motor learning, and change in MEPs with 30 min exercise [129]. The difference between these and the aforementioned results may imply that compensation for the long-term impacts of *BDNF* polymorphisms may have occurred. Some studies suggest that interventions reverting BDNF levels in the motor cortex might prevent the age-related decline in motor function. However, it is still debatable whether such intervention is truly beneficial [117,119,126,127].

Utilizing neuronal plasticity in the motor cortex to improve impaired brain function has been a research topic recently (Table 2). Some neuronal plasticity enhancers have improved motor learning in rodent studies [23,58]. The membrane protein Nogo-A and its receptor (NgR1) are expressed in layer II/III and V pyramidal neurons in the motor cortex. Functional blocking antibodies against Nogo-A or NgR1 enhance LTP in the layer II/III primary motor cortex and increase spine formation in vivo in the layer V neurons. Anti-Nogo-A antibody treatment using an osmotic minipump has improved the success rate in the pellet-reaching task of *rats* [58]. Meanwhile, a small compound edonerpic maleate facilitates synaptic delivery of AMPA receptors, enhances mEPSC amplitudes, and enhances LTP in layer II/III pyramidal neurons of the barrel cortex. This effect is abolished in the absence of whiskers, indicating that the effects of ednerpic maleate are exerted in an experience-dependent manner. The combination of edonerpic maleate administration and rehabilitative training has accelerated the success rate in the pellet-reaching task of the *mice* with cortex cryoinjury [23].

## 5. Conclusions

Many unanswered questions remain about the relationship between age-related alteration of synaptic plasticity and motor function decline in the motor cortex. Nevertheless, evidence from rehabilitation and other studies suggests that enhancing motor function by modulating synaptic plasticity can be an effective intervention for *humans*. The use of brain substances that decrease with age is an important aspect of developing preventive methods against age-related motor decline.

## Figures and Tables

**Figure 1 cells-12-02142-f001:**
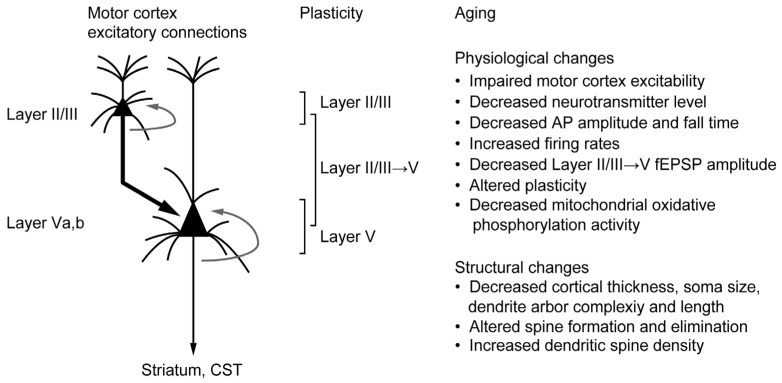
Age-related changes in the motor cortex excitatory connections. Physiological and structural properties change in the motor cortex excitatory connections during physiological aging. These age-related changes cause impairment of motor cortex excitability. The excitatory connection from layers II/III to V is the major intralaminar connection in the motor cortex. Gray arrows from pyramidal neurons in layers II/III or V indicate intralaminar horizontal connections. Abbreviations: CST, corticospinal tract; AP, action potential; fEPSP, field excitatory postsynaptic potential.

**Table 1 cells-12-02142-t001:** Plasticity in the motor cortex.

Species	Age	Motor Learning Task or In Vivo Stimulation	Stimulation Layer/Area	Recording Layer/Neuron	Plasticity Type	Amount of Plasticity	Recording or Measurement Method	References
Female *rats*	Adult	Pellet-reaching task	Layer II/III at 500 μm horizontally from the recording electrode in the M1 forelimb area *	Layer II/III at 200–350 μm below the pial surface in the M1 forelimb area *	LTP	FP amplitude ↑	Field potential recording	[42,43,44]
Male *rats*	Not mentioned	Pellet reaching task	Layer II/III at 3.0 mm lateral to the midline	Layer II/III at 2.0 mm lateral to the midline in the M1 forelimb area *	LTP	fEPSP amplitude ↑	Field potential recording In vivo recording	[45]
Male *rats*	Not mentioned	In vivo white matter stimulation (previously potentiated *rat*)	Layer II/III in the primary motor cortex Layer V in the primary motor cortex	Layer II/III in the primary motor cortex Layer V in the primary motor cortex	FP	FP amplitude ↑ FP amplitude ↔	Field potential recording	[54]
Male *rats*	8–10 weeks	None	Layer II/III at 2–2.5 mm lateral to the midline	Layer II/III at 500 μm lateral to the stimulation electrode in the M1 forelimb area *	LTP	fEPSP amplitude ↓ (D1 or D2 receptor antagonist after LTP induction vs. control)	Field potential recording	[55]
*Mice*	1 month >4 months 1 month	Accelerated rotor rod task		Layer V pyramidal neurons in the M1 forelimb area *	Structural plasticity (2-days trained *mice*) Structural plasticity (After previous 2-days training)	Spine formation ↑ (2-days) Spine formation ↔ (Next 2-days)	In vivo two-photon imaging	[56]
*Mice*	1 month	Pellet reaching task		Layer V pyramidal neurons in the motor cortex	Structural plasticity	Spine formation ↑ Spine elimination ↔	In vivo two-photon imaging	[57]
Male *rats* Male and female *mice*	5–6 weeks 1 month	None None Pellet reaching task	Layer II/III 2–4 mm lateral to the midline	Layer II/III at 500 μm lateral to the stimulating electrode Layer V neurons in the motor cortex Layer II/III pyramidal neurons in the motor cortex	LTP Structural plasticity Structural plasticity	fEPSP amplitude ↑ (anti-Nogo A vs. control) Spine formation ↑ (anti-Nogo A vs. control) Spine density ↑ (sham and anti-Nogo A)	Field potential recording In vivo two-photon imaging	[58]
Male and female PD model *mice*	1–3 months	Dopamine depletion	Superficial layers of the motor cortex	Layer V pyramidal neurons in the M1 forelimb area * Layer V pyramidal neurons 10–100 μm below the cortical surface in the motor cortex	LTP Structural plasticity	EPSC amplitude ↓ (DA depletion vs. control) Spine turnover in the dendritic spine ↑ (DA depletion vs. control)	Whole-cell recording In vivo two-photon imaging	[59]
Male *rats*	10–12 weeks	Pellet reaching task		Layer V neurons	Structural plasticity	Dendritic length ↑ (after 1 month)	Histological analysis	[60]
*Rats*	55–59 days	Pellet reaching task	Entire cortical slice centered over the recorded neurons in the primary motor cortex	Layer V neurons in the caudal forelimb area	Photo-induced EPSC	EPSC amplitude↑ PPR ↔	Whole-cell recording	[47]
Male *Rats*	4 weeks	Accelerated rotor rod task	Layer II/III at 200–300 μm laterally from the recorded neurons in the primary motor cortex	Layer II/III pyramidal neurons in the M1 forelimb area *	mEPSC mIPSC	Amplitude ↑ (1-day and 2-days trained) Frequency ↔ (1-day), ↑ (2-days trained) AMPA/NMDA ↑ (1 day), ↔ (2-days trained) PPR ↔ (1-day), ↓ (2-days trained) Amplitude ↔ (1-day), ↔ (2-days trained) Frequency ↓ (1 day), ↔ (2-days trained) PPR ↑ (1 day), ↔ (2-days trained)	Whole-cell recording	[40,48]
Male and female *mice*	1 month, 4 months	Pellet reaching task		Layer II/III pyramidal neurons Layer V pyramidal neurons	Structural plasticity Structural plasticity	Spine formation and elimination ↔ Spine formation and elimination ↑	In vivo two-photon imaging	[61]
Male *mice*	30–45 days	Repeated tDCS	Layer II/III in the primary motor cortex	Layer II/III at ~200 μm lateral to the stimulation electrode in the primary motor cortex	LTP mEPSC mIPSC Structural plasticity	fEPSP amplitude ↑ PPR (interval: 20 ms) ↓ AMPA/NMDA ratio ↑ Amplitude ↔, Frequency ↑ Amplitude and frequency ↔ Spine density ↑	Field potential recording Whole-cell recording Histological analysis	[49]
Male *mice*	15–18 months	CoQ_10_ suppllementation None	Layer II/III in the radial direction from the recording electrode	Layer V in the primary motor cortex	fEPSP LTP	fEPSP amplitude ↑ (CoQ_10_ middle-aged vs. age-matched control) fEPSP amplitude ↑ (CoQ_10_ during LTP induction vs. age-matched control)	Field potential recording	[50]
Male *rats* Male *mice*	4 weeks 8–10 weeks	Accelerated rotor rod task Accelerated rotor rod task	Layer II/III at 200–300 μm laterally from the recorded neurons in the primary motor cortex	Layer V pyramidal neurons in the M1 forelimb area * Layer V pyramidal neurons in the motor cortex	mEPSC mIPSC Structural plasticity	Amplitude ↔ (1 day), ↑ (2 days trained) Frequency ↔ (1 day), ↑ (2 days trained) AMPA/NMDA ratio ↔ (1 day), ↑ (2 days trained) PPR ↔ (1 day and 2 days trained) Amplitude ↓ (1 day), ↔ (2 days trained) Frequency ↓ (1 day), ↔ (2 days trained) PPR ↑ (1 day), ↔ (2 days trained) Volume of spines ↑	Whole-cell recording In vivo two-photon imaging	[41]

Abbreviations: ↑, increase; ↓, decrease; ↔, no change; the M1 forelimb area *, the region corresponding to forelimb representation in the primary motor cortex * [62]; FP, field potential; fEPSP, field excitatory postsynaptic potential; mEPSC, miniature excitatory postsynaptic current; mIPSC, miniature inhibitory postsynaptic current; PPR, paired-pulse ratio; PD, Parkinson’s disease model.

**Table 2 cells-12-02142-t002:** Intervention methods for targeting CNS for the decline of motor function.

Intervention Type	Administration Method	Species	Age	Effects on Motor Function	Cell or Brain Region	Target Mechanism	Commercial Availability	References
Nanomicellar formulation of CoQ_10_ supplementation	Oral	Male MPTP treated-*mice*	8–10 weeks	Decrease of hindlimb faults number during the beam walk test	Substantia nigra, Striatum	Neuroprotection, Astrocytic activation in the midbrain	Yes	[112]
Water-soluble nano formula-type CoQ_10_ (ubiquinone) supplementation	Oral	Male and female *mice*	15 months	Improvement of the pole test latency	Motor cortex	Brain mitochondrial oxidative phosphorylation dysfunction	Yes	[10]
	Oral	Male *mice*	15–18 months	Improvement of the pole test latency	Primary motor cortex	Age-related decline of neuronal activity in layer V in the primary motor cortex, Brain mitochondrial oxidative phosphorylation dysfunction	Yes	[50]
High-dose CoQ_10_ (ubiquinol) supplementation	Oral	Male and female multiple-system atrophy patients	Median age 61.0 years	Improvement of SARA score and time required to walk 10 m	Cerebellum, Motor cortex, Putamen	CoQ_10_ deficiency (*COQ2* mutation)	Yes	[108]
Anti-Nogo-A antibodies treatment	Continuous intrathecal infusion	Male *rats*	5–6 weeks	Increase in the success rate of the pellet-reaching task	Layer II/III and V neurons in the motor cortex	Spine formation, Spine density modulation upon motor learning in the primary motor cortex	Limited	[58]
Combination of edonerpic maleate administration and rehabilitative training	Oral	Male *mice* after motor cortex cryoinjured Male *monkey* after motor cortex cryoinjured	5–13 weeks 5 or 6 years	Facilitation of recovery from injury of the motor cortex (the food-reaching task performance)	Layer V pyramidal neurons in the motor cortex	Experience-dependent synaptic AMPA receptor delivery	Limited	[23]
Exercise (Increase in activity-dependent BDNF secretion/TrkB phosphorylation)		*Mice*/*Human*	Not mentioned	Improvement of motor learning	Layer II/III neurons in the motor cortex	Activity-dependent BDNF secretion/TrkB phosphorylation, BDNF-mediated synaptic plasticity (LTP)	Not applicable	[70,117,119,120,126,127]

Abbreviations: CNS, central nervous system; CoQ_10_, coenzyme Q_10_; BDNF, brain-derived neurotrophic factor; TrkB, BDNF tyrosine receptor kinase B; SARA score, Scale for the Assessment and Rating of Ataxia score.

## Data Availability

Not applicable.

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
