# Peer review of "Neuronal Plasticity and Age-Related Functional Decline in the Motor Cortex"

_cells, 2023, doi:10.3390/cells12172142_

Round 1

Reviewer 1 Report

Dear Authors

The review is well written and characterize precisely the field of motor cortex and sinaptic plasticity.

the table are very explicative

a little paragraph related to spine density and morphology is well acceptable.

thanks for comprehension

Author Response

Thank you very much for the positive comments.

“a little paragraph related to spine density and morphology is well acceptable.”

Thank you for the comment. In response to Reviewer1’s comment, we revised the sentence in lines 173 to 176 as follows, “Repeated in vivo imaging of the same dendrites has revealed that dendritic spine density increases and long-term spine survival decreases in layer V pyramidal neurons of aged mice (20 to 22 months old) compared to young mice (3 to 4 months old) [81].” In addition, we added a new Figure 1 and described the spine density in the “Aging” section.

The references in Table 1 describe multiple aspects of spine dynamics, including formation, elimination, density, and volume, but not all references report spine density. We have described these references about dendritic spines in paragraphs 2.2.3, 3.1, and the last paragraph on page 11. However, we feel that there is not sufficient material for an independent paragraph.

Reviewer 2 Report

The review “Neuronal plasticity and age-related functional decline in the motor cortex” by Ritsuko Inoue and Hiroshi Nishimune, addresses the causes of motor decline that accompanies aging. This is accomplished by first compiling the publications that study plasticity in the motor cortex. Both electrophysiologically induced synaptic plasticity (i.e., LTP) and learning-induced plasticity in the motor cortex. To then focus on the changes observed in the motor cortex with age. Finally, it recapitulates the interventions that compensate or ameliorate age related motor decline.

This work makes a very complete review of the plasticity of the motor cortex. A strength of this review is that it explores the possible origins of motor decline associated with aging, relating them to the changes in plasticity that occur with age. The topic of the review is relevant to the area of ​​knowledge and fills the gap in knowledge of the role of motor cortex in age-associated motor impairment.

The review is clear and presented in a well-structured manner. To my knowledge the references are relevant, recent, and extensive.

The manuscript concludes by reviewing interventions for age-related declines of motor function. I think the publication would greatly benefit from a general conclusions section.

Author Response

Thank you very much for the recognition and positive comments on our manuscript.

Reviewer 3 Report

The Review is well-described and presented well.  Writing is very clear and easy to comprehend despite the complexity of the subject material.

Just some minor suggestions for the authors to consider are listed below.

Line 51: “The hypoexcitability in the motor cortex correlates with behavioral impairments in chronic obstructive pulmonary disease (COPD) and amyotrophic lateral sclerosis (ALS) patients [19,20]. Therefore, alteration of motor cortex function may be part of the underlying mechanism for the age-related decline of motor function, especially in the middle-aged stage.” – Are behavioral impairments only seen in patients with COPD and ALS? If so, how can the conclusion be drawn that this alteration of motor cortex function is a mechanism for overall motor function decline?

Line 89: “Changes in motor cortex excitability have functional relation with LTP.” – Change to a functional relationship

Table 1 “500 μM” – Believe this may be a typo (μm instead) ---- under “Male rats”

Line 146 : “Physiological aging causes cortical atrophy, impaired excitability, and decreased neurotransmitter levels in the human motor cortex” – Identical to sentence in the introduction, may want to rephrase or change

Line 253: “Stimulation of superficial layers of the motor cortex using a concentric bipolar electrode induces LTP in layer V pyramidal neurons in the M1 forelimb area” – There is a font size change here.  

none

Author Response

Thank you very much for the positive comments.

Just some minor suggestions for the authors to consider are listed below.

Line 51: “The hypoexcitability in the motor cortex correlates with behavioral impairments in chronic obstructive pulmonary disease (COPD) and amyotrophic lateral sclerosis (ALS) patients [19,20]. Therefore, alteration of motor cortex function may be part of the underlying mechanism for the age-related decline of motor function, especially in the middle-aged stage.” – Are behavioral impairments only seen in patients with COPD and ALS? If so, how can the conclusion be drawn that this alteration of motor cortex function is a mechanism for overall motor function decline?

[Response] Thank you for the comment. Motor function decline can be seen in other neurological diseases, including preclinical Alzheimer’s disease and Parkinsonian syndromes. These neurological diseases affect other parts of the central nervous system and not necessarily the motor cortex. Therefore, the motor function decline may be caused by multiple mechanisms. However, the motor cortex is the commanding center for voluntary movement. Therefore, we reasoned that if the function of the commanding center is altered, it is likely to take part in the mechanism of the motor function decline. We do not intend to describe that the “hypoexcitability in the motor cortex” is “the only mechanism” or “for overall motor function” decline. Therefore, we described it as “correlates with” and “may be part of the underlying mechanism.”

Line 89: “Changes in motor cortex excitability have functional relation with LTP.” – Change to a functional relationship

 [Response] Thank you for the comment. We have revised the text as specified, “a functional relationship”. 

Table 1 “500 μM” – Believe this may be a typo (μm instead) ---- under “Male rats”

[Response] Thank you for pointing out our error. We have corrected Table 1 as “500μm”.

Line 146 : “Physiological aging causes cortical atrophy, impaired excitability, and decreased neurotransmitter levels in the human motor cortex” – Identical to sentence in the introduction, may want to rephrase or change

 [Response] Thank you for the comment. We have revised the text as, “Physiological aging induces changes in the human motor cortex, including cortical atrophy, impaired excitability, and decreased neurotransmitter levels [15,16,73,74].”

Line 253: “Stimulation of superficial layers of the motor cortex using a concentric bipolar electrode induces LTP in layer V pyramidal neurons in the M1 forelimb area” – There is a font size change here.  

 [Response] Thank you for the comment. We have corrected the font size of the text.

Reviewer 4 Report

Ther manuscript entitled "Neuronal plasticity and age-related functional decline in 2 the motor cortex" by Ritsuko Inoue and Hiroshi Nishimune is a well-writen and well-curated review on plasticity and age-related decline in the motor cortex. The authors have also included an Interesting and exciting section on possible therapeutic approaches to enhance motor function.

My only comment to the authors relates to Tables 1 and 2; these are very informative but I think that an additional Figure/Scheme summarizing the networks and anatomical details that the authors refer to in the text might be more helpful, particularly for the non-specialist reader.

Author Response

Thank you for the comment. We have added a new Figure1 in the manuscript.

Round 2

Reviewer 2 Report

I am satisfied with the revised version of the manuscript